# Hepatocellular Carcinoma: Surveillance, Diagnosis, Evaluation and Management

**DOI:** 10.3390/cancers15215118

**Published:** 2023-10-24

**Authors:** Jessica Elderkin, Najeeb Al Hallak, Asfar S. Azmi, Hussein Aoun, Jeffrey Critchfield, Miguel Tobon, Eliza W. Beal

**Affiliations:** 1Wayne State University School of Medicine, Detroit, MI 48201, USA; hn2433@wayne.edu; 2Department of Oncology, Karmanos Cancer Institute, Wayne State University School of Medicine, Detroit, MI 48201, USA; alhallakm@karmanos.org (N.A.H.); azmia@karmanos.org (A.S.A.); 3Department of Radiology, Karmanos Cancer Institute, Wayne State University School of Medicine, Detroit, MI 48201, USA; aounh@karmanos.org (H.A.); aa8442@wayne.edu (J.C.); 4Department of Surgery, Karmanos Cancer Institute, Wayne State University School of Medicine, Detroit, MI 48201, USA; mtobon@med.wayne.edu

**Keywords:** hepatocellular carcinoma, systemic therapy, locoregional therapy

## Abstract

**Simple Summary:**

Hepatocellular carcinoma (HCC) is the most common primary liver cancer. High-risk patients should undergo semi-annual surveillance. Management of HCC is complex and requires multidisciplinary team evaluation and the consideration of patient’s goals of care. Early HCC is best managed by curative-intent treatment with liver resection, transplantation, or ablation. Intermediate-stage disease may be treated with transarterial therapies. Advanced-stage disease should be treated with systemic therapy. There are many therapies and therapy combinations currently under investigation.

**Abstract:**

Hepatocellular carcinoma (HCC) ranks fourth in cancer-related deaths worldwide. Semiannual surveillance of the disease for patients with cirrhosis or hepatitis B virus allows for early detection with more favorable outcomes. The current underuse of surveillance programs demonstrates the need for intervention at both the patient and provider level. Mail outreach along with navigation provision has proven to increase surveillance follow-up in patients, while provider-targeted electronic medical record reminders and compliance reports have increased provider awareness of HCC surveillance. Imaging is the primary mode of diagnosis in HCC with The Liver Imaging Reporting and Data System (LI-RADS) being a widely accepted comprehensive system that standardizes the reporting and data collection for HCC. The management of HCC is complex and requires multidisciplinary team evaluation of each patient based on their preference, the state of the disease, and the available medical and surgical interventions. Staging systems are useful in determining the appropriate intervention for HCC. Early-stage HCC is best managed by curative treatment modalities, such as liver resection, transplant, or ablation. For intermediate stages of the disease, transarterial local regional therapies can be applied. Advanced stages of the disease are treated with systemic therapies, for which there have been recent advances with new drug combinations. Previously sorafenib was the mainstay systemic treatment, but the recent introduction of atezolizumab plus bevacizumab proves to have a greater impact on overall survival. Although there is a current lack of improved outcomes in Phase III trials, neoadjuvant therapies are a potential avenue for HCC management in the future.

## 1. Introduction

Hepatocellular carcinoma (HCC) is the dominant type of primary liver cancer, and the fourth most common cause of cancer-related deaths worldwide. This is the case despite the development and implementation of hepatitis C treatment and hepatitis B vaccination and antiviral therapy [1,2,3]. Most cases of HCC worldwide occur in patients with chronic liver disease from hepatitis B or C infections [4]. Cirrhosis is the leading risk factor and cause of mortality in patients with HCC [5,6]. The preventability of liver cirrhosis by way of antivirals has led the World Health Organization to establish an agenda to eliminate the hepatitis B virus (HBV) by 2023 through vaccinations, diagnostic tests, and education campaigns [7,8]. Unfortunately, there is an increasing incidence in previously lower-risk areas, which may be associated with a growing prevalence of metabolic diseases, such as non-alcoholic steatohepatitis (NASH) and non-alcoholic fatty liver disease (NAFLD) [5,9,10,11]. Many studies have shown that over the last few decades, the proportion of the population that is overweight or obese has risen with an associated increase in HCC [12,13,14,15]. When acting synergistically with other lifestyle metabolic disease risk factors (e.g., diabetes mellitus, tobacco, or alcohol consumption), the risk of HCC increases [16,17].

The implementation of surveillance programs for chronic liver disease, cirrhosis, and HCC allows for earlier tumor detection, curative treatment, and longer overall survival, as compared to when patients present symptomatically with the advanced disease [18,19]. The management of HCC requires a multidisciplinary team and can be complex, depending on the stage of the disease at presentation. For early-stage HCC, curative treatments may be offered, such as surgical resection, ablation, or liver transplantation [20]. Transarterial locoregional therapies are recommended for patients presenting with intermediate disease stages [21]. Systemic therapies are indicated in patients with the advanced disease [22]. This review will discuss surveillance for HCC in high-risk patients, as well as the diagnosis, evaluation, and management of HCC.

## 2. Surveillance

The goal of surveillance programs is to identify early-stage HCC in at-risk patients (cirrhosis from any etiology, HBV in the absence of cirrhosis, and HCV in the absence of cirrhosis), when curative-intent treatment is an option [23]. The American Association for the Study of Liver Diseases (AASLD), Asian Pacific Association for the Study of the Liver (APASL), the National Comprehensive Cancer Network (NCCN), and the European Association for the Study of the Liver (EASL) all recommend surveillance every six months for these high-risk patients [24,25,26,27]. Abdominal ultrasonography (US) is the recommended surveillance modality [28]. When used in combination with the biomarker alpha-fetoprotein (AFP), the sensitivity of early HCC surveillance significantly increases [29]. One study found that the sensitivity of US and AFP in detecting HCC were 44% and 66% with specificities of 92% and 91%, respectively. Sensitivity increased to 90% when US and AFP were used in combination, with a slight loss in specificity to 83% [30]. The use of AFP can be controversial due to the loss in specificity. Therefore, not all international guidelines recommend AFP for surveillance [28]. Obesity, non-alcoholic steatohepatitis (NASH) and non-alcoholic fatty liver disease (NAFLD) are also risk factors for HCC, although surveillance recommendations for these patient populations have not been included in current guidelines [31]. In obese patients, bariatric surgery has been shown to decrease the risk of HCC [32].

Although the HCC surveillance of patients with cirrhosis has been found to increase survival, surveillance is underused in clinical practice [33]. Barriers have been identified at the patient level, provider level, and system level. Identified patient-level barriers include financial constraints, lack of awareness of surveillance recommendations, and scheduling issues [34]. At the provider level, there is a lack of clinician orders for surveillance and cirrhosis unrecognition [35]. Primary care providers (PCPs) care for a large proportion of patients with cirrhosis. In a study examining practice patterns of PCPs, it was demonstrated that PCPs have misconceptions about the tests that detect HCC and are not fully acquainted with surveillance guidelines [36].

Due to the continued underuse of surveillance programs, it is recommended that interventions be implemented to improve HCC surveillance rates [35,37]. A variety of interventions have been evaluated. Patient-level interventions commonly involve patient education on the importance of screening, assistance with scheduling, and programs for transportation to screening appointments [34,38,39]. Provider-level interventions include adherence to screening guidelines, reminders via electronic medical records, and compliance reports [40,41]. On the macro-level, initiatives that aim to decrease the cost of screening and increase the accessibility could positively impact the surveillance of HCC [34].

## 3. Diagnosis

Imaging plays a critical role in the diagnosis of HCC (Figure 1), making it unique from most other solid tumors that require biopsy confirmation. HCC lesions have a characteristic appearance in US, contrast-enhanced US, computed tomography (CT), and magnetic resonance imaging (MRI), allowing for non-invasive diagnosis and characterization [37].

The Liver Imaging Reporting and Data System (LI-RADS, Table 1) is a comprehensive system that standardizes the terminology, technique, interpretation, reporting, and data collection of image-based liver observations in patients with cirrhosis, chronic HBV, and patients with current or prior HCC, as well as high-risk non-cirrhotic patients [42,43]. Exclusion criteria for LI-RADS consist of patients with cirrhosis due to vascular disorders or congenital hepatic fibrosis [44]. LI-RADS is supported by both the American College of Radiology (ACR) as well as the American Association for the Study of Liver Disease (AASLD) and is a widely accepted system for the diagnostic imaging techniques in HCC [45]. LI-RADS has four different algorithms that can be utilized depending on the clinical context: the US LI-RADS modality for surveillance, CT/MRI LI-RADS for diagnosis and staging, contrast-material-enhanced US LI-RADS for diagnosis, and treatment response LI-RADS for monitoring the response to local-regional therapies [46].

Per EASL and AASLD guidelines, the diagnosis of HCC in cirrhotic patients should be based on non-invasive criteria and/or pathology, while in non-cirrhotic patients, it should be confirmed by pathology. While US is the primary surveillance modality, in the case of diagnosis, CT or MRI should be used due to higher sensitivity. These organizations also advise against routine biopsy, unless a timely diagnosis is required, the image-based diagnosis is inconclusive, or for the diagnosis of HCC in patients without cirrhosis [24,27].

## 4. Staging

HCC is different from other malignancies in that liver function plays an important role in prognosis along with tumor stage [47,48,49,50]. The conventional tumor-node-metastasis (TNM) staging system (Table 2) has clinical limitations for HCC due to only considering the tumor stage for prognosis, and it has been shown that the TNM staging system is a poor prognostic predictor for surgical resection and transplantation in patients with HCC [48,49,50]. The clinical usefulness of including liver function in prognoses is the reason that it has been included in most HCC staging systems, including the Barcelona Clinic Liver Cancer (BCLC) system, Japanese integrated staging (JIS) system, Chinese University Prognostic Index (CUPI), and Cancer of the Liver Italian Program (CLIP) [51,52,53,54]. The BCLC staging system was described in 1999, with its most recent update in 2022, and stratifies HCC based on the extent of the primary tumor, performance status (PS), vascular invasion, extrahepatic spread, and the amount of underlying liver function. Patients are placed into one of five categories depending on the aforementioned characteristics, and a corresponding therapy is recommended [51,55]. The JIS was reported in 2003 and utilizes the TNM staging by the Liver Cancer Study Group of Japan (LCSGJ) criteria along with the Child-Pugh. This staging system is not utilized in Western countries [52,55]. The CUPI was published in 2002 and stratifies patients by total bilirubin, alkaline phosphatase, alpha-fetoprotein, and asymptomatic disease at presentation, as well as the TNM system [53,55]. CLIP was created in 1998 and involves tumor morphology, serum alpha-fetoprotein levels, and portal vein thrombosis in concert with Child-Pugh [54,55]. Of these staging systems, the BCLC is most commonly used in Western countries [56].

The BCLC staging system was originally developed for the stratification of patients with cirrhosis and HCC with their respective treatment recommendations, and is now also applied in patients without cirrhosis (Figure 2) [58]. Although there is no consensus globally as to which staging system to use, the BCLC staging system is endorsed by EASL and AASLD and has become the reference staging system in Western countries [48,59]. The BCLC staging system categorizes HCC into five stages (0-C) based on tumor burden, liver function, and the Eastern Cooperative Oncology Group’s Performance Status Scale. Liver dysfunction is evaluated by evidence of jaundice, ascites, and encephalopathy. The five different categories in which HCC can be placed correspond to treatment recommendations, such as ablation, resection, transplant, transarterial chemoembolization (TACE), systemic treatment, and best supportive care (BSC). The updated 2022 BCLC strategy includes additional stratification in the BCLC-B group based on tumor burden and liver function, as well as an option for liver transplant in the BCLC-B group after meeting the Extended Liver Transplant criteria [60]. There is also the inclusion of tremelimumab-durvalumab in addition to the previous atezolizumab with bevacizumab (Atezo-Bev) immunotherapy as a first-line treatment option, and the idea of treatment stage migration is now incorporated into the BCLC model [51].

## 5. Curative-Intent Therapies

The three curative-intent treatment modalities are resection, transplant, and local ablation, and should be utilized whenever feasible [27]. The decision between liver transplant and resection in patients suitable for curative therapies can be a difficult one. While a liver transplant may take care of both a cirrhotic liver and HCC, it also relies on the availability of a donor organ and requires lifestyle changes owing to lifelong immunosuppressive therapy. Although surgical resection does not require the availability of an organ, it does have the potential of liver decompensation and a high risk of recurrence due to undetectable microscopic metastases that remain unresected [61,62,63].

Surgical resection is the treatment of choice in non-cirrhotic patients with a non-fibrotic liver, no underlying hepatitis, and no vascular invasion [64]. EASL and AASLD recommend resection in patients with cirrhosis who are in early-stage HCC with localized tumors, well-compensated cirrhosis, and without clinically significant portal hypertension [24,27,65]. There are two techniques of resection: anatomic resection (AR) and non-anatomic resection (NAR). In AR, the hepatic segment of the liver containing the tumor is removed along with all tumor-bearing tributaries. NAR is a parenchyma-sparing technique. AR is superior to NAR in terms of disease-free survival (DFS) and overall survival (OS) among patients undergoing resection in HCC [66,67,68]. Minimally invasive liver resection may be recommended over open resection in eligible patients as it correlates with shorter hospital stays and reduced post-operative complications [69,70]. In the last two decades, the availability of the robotic platform has created an increasing trend toward the minimally invasive surgical resection of HCC as large as hemihepatectomies [71,72,73,74].

Liver transplantation (LT) is indicated for patients who are not candidates for surgical resection but are within the Milan criteria. The Milan criteria were introduced after a 1996 study defined selection criteria for transplant patients that led to superior survival, and it has since become the standard of care for early HCC [75,76]. Macrovascular invasion and extrahepatic metastases are definite contraindications for LT [77,78]. For some patients outside of the Milan criteria, there is the potential of becoming a candidate for LT after successful downstaging using locoregional therapies. A group at the University of California, San Francisco, conducted a 12-year study in which they performed LT on patients outside of the Milan criteria and found that patients with solitary tumors < 6.5 cm, or <3 nodules with the greatest lesion being under 4.5 cm and a total lesion diameter of <8 cm, had survival rates at the one and five year marks of 90% and 75.3%, respectively [79]. The extended criteria have been tested and shown to have comparable outcomes to LTs within the Milan criteria [80]. Additionally, successful downstaging of HCC to fit into the Milan criteria has been shown to produce comparable post-operative outcomes as LT to patients initially presenting within the Milan criteria [81,82]. The consideration of LT in patients originally outside of the Milan criteria after down-staging has been incorporated into both the ASLD and EASL guidelines [24,27]. As mentioned previously, a barrier to LT is the availability of donor organs, for which living donor liver transplantation (LDLT) has presented a partial solution, reducing waitlist mortality with equivalent, and in some cases, superior survival [83,84,85]. The United States only utilizes LDLT in 1.5% of all LTs. However, LDLT is used more frequently in Asian countries and other countries with graft scarcity [86]. The model for end-stage liver disease (MELD) score predicts the survival of patients with advanced liver disease and its major use is the determination of the allocation of organs for liver transplants [87]. For non-cirrhotic or cirrhotic but low MELD score patients, it is difficult to find available organs due to a high median MELD score of 33 in the United States [88]. Options other than LDLT include expanding the criteria for patients older than 60, cold ischemia time greater than 8 h, marginal grafts with macrosteatosis over 30%, partial/split donors, elevated liver function tests, and hepatitis C donors. Additionally, the use of normothermic machine perfusion to evaluate the function of more marginal grafts and donation after circulatory death may improve graft access [89,90,91,92].

## 6. Locoregional Therapies

Locoregional therapies are commonly used in the management of HCC, comprising treatment in 50–60% of cases. The goal of locoregional therapies is to reduce tumor viability, progression, and extend overall survival in patients with HCC [93]. When HCC patients are not operative candidates, ablation offers a curative-intent therapy option. Ablation is achieved by percutaneous ethanol injection (PEI), radiofrequency ablation (RFA), microwave ablation (MWA), cryoablation, and high-intensity focused ultrasound (HIFU) ablation modalities, of which the RFA modality is the most used globally [94]. RFA, MWA, cryoablation, and HIFU manipulate temperature to induce cell necrosis, while PEI introduces chemical agents [95]. Advantages of MWA over RFA include a greater extent of cellular necrosis, reduced procedural time, higher temperatures applied to target lesions, and reduced heat-sink effect [96,97]. As a result of excellent outcomes in a minimally invasive setting, ablation is equivalently as effective as resection, and is sometimes recommended as first-line therapy in very early stages of HCC by the AASLD and EASL [24,27,98]. Local ablative therapies are also used in the context of bridging therapies for LT, to enable downsizing of the lesion, reduce drop-out from the waiting list, and improve prognosis after LT [99,100,101,102].

Other methods take advantage of the hypervascularity of tumors and their unique reliance on blood supply from the hepatic artery, whereas hepatic parenchyma blood supply is largely from the portal vein, allowing transarterial therapies to be target selective [103]. Transarterial chemoembolization (TACE) is the standard of care for the intermediate stages of HCC and involves both creating local hypoxia and intraarterial infusion of a chemotherapy agent [103,104]. Due to the intermediate stage of HCC encompassing tumors with different tumor burdens, liver functions, and disease etiologies, the benefits of TACE are not the same for every patient. However, the intermediate class, as a whole, has improved survival with this therapy [105]. TACE is contraindicated in patients with advanced cirrhosis or poor functional status. Additionally, main portal vein lesions, biliary obstruction, and hepatic encephalopathy are considered relative contraindications [103]. Transarterial radioembolization (TARE) with yttrium-90 (Y-90) has recently gained traction in the treatment of HCC, and has entered the treatment algorithm in BCLC and AASLD for intermediate stages of HCC [56,106]. Comparing TACE versus TARE has yielded mixed results with some studies showing similar survival outcomes between the two, and others favor TARE over TACE in downstaging HCC, longer time to progression, and quality of life [107,108,109,110]. Hepatic artery infusion chemotherapy (HAIC) is an alternative treatment modality that provides increasing therapeutic efficacy by delivering drugs directly into tumor feeding vessels, minimizing systemic toxicities [111]. HAIC shows favorable outcomes over systemic therapy with sorafenib for patients without macroscopic vascular invasion. Inversely, systemic therapy is recommended for patients with macrovascular invasion of HCC [112]. Stereotactic body radiation therapy (SBRT) is a modality of treatment that can be used as an adjunct to other treatments or alone in cases where other local therapies are unsuitable [113,114,115]. Randomized control trials demonstrated the superior ability of the long-term local control of HCC lesions with external beam radiation therapy over RFA, although phase III trials comparing SBRT with other treatment modalities are ongoing [114,115].

## 7. Systemic Therapies

The advent of novel agents for systemic therapy in HCC gives promise to treatment in advanced stages of disease (Table 3). In 2007, sorafenib, an oral multikinase inhibitor, was approved and considered the standard of care in the management of unresectable HCC [116]. In more recent years, studies such as REFLECT compared lenvatinib to sorafenib, another multikinase inhibitor, and showed lenvatinib to be non-inferior to sorafenib, making it an additional first line option [117,118]. More recently, the combination of anti-PD-L1 antibody atezolizumab and anti-VEGF antibody bevacizumab was demonstrated to be superior in terms of OS and PFS in comparison to sorafenib in the IMbrave150 trial [119,120,121]. The atezolizumab plus bevacizumab (Atezo-Bev) combination was approved by the FDA in 2020 and is an additional first-line option in unresectable HCC [122]. Later in 2022, the FDA approved the first dual immunotherapy regimen (durvalumab plus tremelimumab) as another first-line systemic treatment for HCC based off the HIMALAYA phase III clinical trial, which showed the combination to be superior to sorafenib [123,124]. This regimen was effective in all subtypes (viral and non-viral). Given the fact that durvalumab-tremelimumab has no anti-VEGF component, the regimen is being favored by clinicians for those patients with a prior history of esophageal variceal bleed or with poorly controlled high blood pressure. There are several ongoing clinical trials for systemic therapies for HCC (Table 4) [125].

## 8. Neoadjuvant Therapies

Neoadjuvant therapies involve local or systemic treatment applied prior to the surgical treatment of HCC to prevent dropouts during LT waiting periods, to shrink tumors to meet LT criteria, to increase the liver resection rates, and to prevent recurrence [147]. This therapy consists of radiation therapy, systemic therapy, immunotherapy, and hormone therapy [148]. TACE has been used to downstage unresectable HCC for resection and to downstage patients outside of Milan LT criteria into transplant candidates. Similarly, TARE utilizes arterially directed therapy to successfully downstage HCC prior to resection or LT [149]. There is compelling evidence that neoadjuvant immunotherapy and chemotherapy provide antitumor effects against cancer cells prior to ablation or surgery [150]. The benefit of immunotherapy stems from improving T cell priming to generate a stronger systemic immune response to facilitate subsequent resection or surgery and prevent recurrence [151]. There are several early-phase clinical trials investigating the efficacy of neoadjuvant immunotherapy, such as CaboNivo, the combination of cabozantinib and nivolumab [152,153]. Although neoadjuvant therapies have not been added to current HCC management due to lacking Phase III trials data, it is a promising avenue for future HCC treatments [154].

## 9. Adjuvant Therapies

There is no currently approved adjuvant therapy after curative-intent treatment for HCC. The efforts to find an adjuvant therapy to reduce the risk of recurrence and prolong survival are still ongoing but none have been significant enough to be implemented in the guidelines. Interferon was tested as an adjuvant therapy in viral-related HCC in the Asian population with promising results showing a survival benefit and reduced recurrence rate when compared to controls [155,156]. The STORM phase III clinical trial was performed to assess the efficacy of sorafenib versus placebo in adjuvant treatment after surgical resection or local ablation. This trial failed to demonstrate a significant improvement in RFS or OS in patients treated with sorafenib versus placebo, which could be partially explained by a high discontinuation rate due to adverse events (24% in the sorafenib treatment group) [157]. IMbrave150 phase III clinical trial compared the combination of atezolizumab and bevacizumab (Atezo-Bev), and PD-1L inhibitor and VEGF inhibitor respectively, to sorafenib post HCC resection/ablation in a high-risk-for-recurrence population (tumor greater than five centimeters, vascular invasion or high-grade pathology). The preliminary results showed improvement in OS and PFS and were confirmed later in another study with Atezo-Bev combined therapy showing a 5–8-months-greater median OS when compared with sorafenib [120,121,158]. Two phase III trials are investigating the role of PD-1 antibodies in the setting of adjuvant therapy after curative resection or ablation, nivolumab (CheckMate 9DX: NCT03383458) and pembrolizumab (KEYNOTE-937: NCT03867084) [159,160]. Postoperative intraarterial injection of 131-iodine-labeled lipiodol into the hepatic artery has been proposed as an adjuvant therapy after surgical resection [161]. The 131-iodine-labeled lipiodol has been shown to benefit disease-free survival and OS in a randomized control trial, although the difference between treatment and control became insignificant at 8 years of follow-up [162]. In China, transarterial chemo embolization is used widely after HCC resection and has potential benefits for patients with increased risk factors such as vascular invasion or multiple nodules [163,164]. However, the studies in this settings are not large and did not include western population and hence adjuvant TACE is not recommended from our view. More randomized studies with long-term follow-up are needed to determine the actual benefits of all these adjuvant therapies in the curatively treated HCC.

## 10. Imaging Response Criteria

The significant role of locoregional therapies in the treatment of HCC requires a method of determining the response of disease to treatment (Table 5). The World Health Organization (WHO) was the first to publish defined standardizations to determine the response radiologically in 1981, followed by Response Evaluation Criteria in Solid Tumors (RECIST) in 2000 [165,166]. Both the WHO and RECIST define response by anatomic size, estimated based on the longest diameter of the lesion, and lesion changes. One of the limitations of these criteria is the lack of sensitivity toward newer molecular targeted therapies that work by inducing tumor necrosis, which may not cause lesion shrinkage initially [167]. EASL proposed a change to the criterion, taking into consideration treatment-induced tumor necrosis as a response [168]. This amended criterion, which considers tumor necrosis, was supported by the AASLD practice guidelines in 2005 [169]. Since then, AASLD-JNCI (Journal of the National Cancer Institute) modified the assessment of the treatment response of RECIST by proposing measurement of only the viable portion of the target lesion instead of the whole lesions. This criterion is termed modified RECIST (mRECIST) [170]. Quantitative EASL (qEASL) and volume RECIST (vRECIST) both acknowledge the three-dimensional volumetric enhancement of tumors [167,169].

## 11. Conclusions

Early detection followed by surgical resection or transplant offers the best outcomes for patients with HCC. The implementation and execution of surveillance programs are crucial to detect HCC before advanced stages of the disease. For intermediate HCC, locoregional therapies such as TACE or TARE allow for targeted treatment of the lesion without systemic side effects. Systemic therapies are reserved for the cases of advanced unresectable HCC. Combined approaches are a more recent addition to the treatment of HCC as are neoadjuvant and adjuvant therapies, which allow downstaging of the disease prior to transplant or resection and help to prevent recurrence.

## Figures and Tables

**Figure 1 cancers-15-05118-f001:**
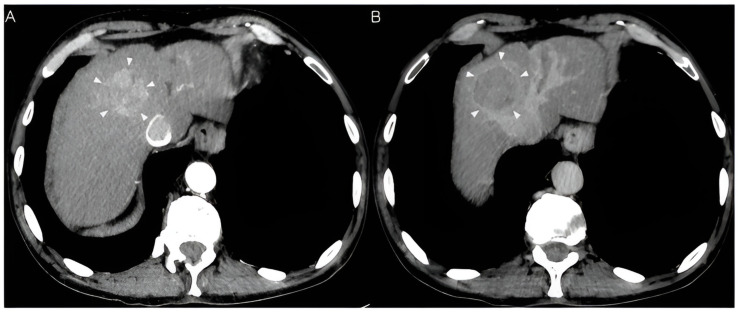
Image (**A**): axial IV enhanced arterial phase CT image demonstrates an enhancing hepatic mass (arrowheads) within segment 4A with early arterial enhancement. Image (**B**): axial IV enhanced delayed phase CT image demonstrates the contrast washout of the mass (arrowheads) relative to normal liver parenchyma, which is classic of hepatocellular carcinoma.

**Figure 2 cancers-15-05118-f002:**
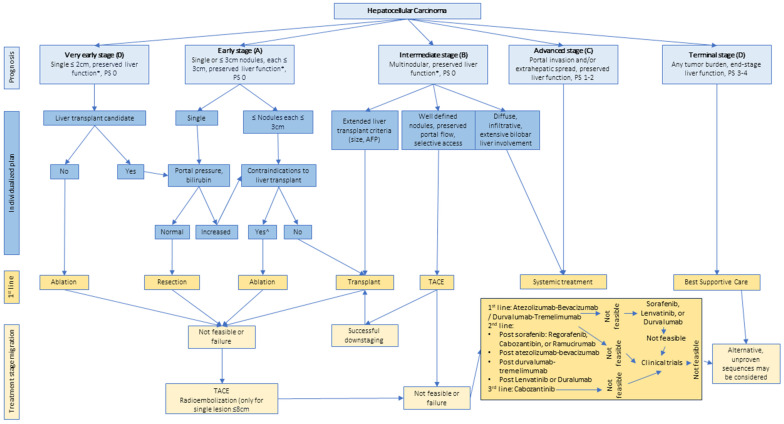
2022 Barcelona Clinic Liver Cancer (BCLC) staging and treatment algorithm. * Except for those with tumor burden acceptable for transplant. ^ Resection may be considered in patients with single peripheral HCC with adequate liver volume.

**Table 1 cancers-15-05118-t001:** Liver lesion characterization for HCC using Liver Imaging Reporting and Data System (LI-RADS) categorization [42,43].

Diagnostic Category	CT/MRI Criteria (Additional Major Features: Enhancing Capsule, Nonperipheral Washout, Threshold Growth)	Management Recommendation
LR-NC: noncategorizable	Cannot be categorized due to image degradation or omission	Repeat or alternative diagnostic imaging in ≤3 months
LR-1: definitely benign	Simple cyst, punctate perfusion alteration, focal fatty deposition/sparing, hemangioma, hypertrophic pseudomass, confluent fibrosis or focal scar No enhancement in both late arterial, portal venous and delayed phase	Return to surveillance in 6 months
LR-2: probably benign, 16% are HCC, 18% are malignant	Distinct nodules <20 mm with no additional major features CT images show lesions with enhancement that follows the blood pool (hemangioma)	Return to surveillance in 6 months OR consider repeat diagnostic imaging in ≤6 months
LR-3: intermediate probability 37% are HCC, 39% are malignant	Nonrim APHE AND ≤ 20 mm with no additional major features No APHE AND <20 mm with ≤1 additional major feature OR ≥20 mm with no additional major features	Repeat or alternative diagnostic imaging in 3–6 months
LR-4: probably HCC, 74% are HCC, 81% are malignant	Nonrim APHE AND: <10 mm with ≥1 additional major feature OR 10–19 mm with enhancing capsule appearance and no other major features OR ≥20 mm with no additional major features No APHE AND: <20 mm with 2 additional major features OR ≥20 mm with ≥1 additional major feature	Multidisciplinary discussion for tailored workup May include biopsy
LR-5: definitely HCC	Nonrim APHE AND: 10–19 mm with nonperipheral washout or threshold growth ≥20 mm with ≥1 additional major feature	HCC confirmed Multidisciplinary discussion for consensus management
LR-M: probably or definitely malignant, not specific for HCC	Targetoid mass: Rim APHE Peripheral washout Delayed central enhancement Targetoid diffusion restriction Targetoid transitional phase or hepatobiliary phase signal intensity Nontargetoid mass not meeting LR-5 criteria AND no TIV, with ≥1 of the following Infiltrative appearance Marked diffusion restriction Necrosis or severe ischemia Other feature suggesting non-HCC malignancy	Multidisciplinary discussion for tailored workup Often includes biopsy
LR-TIV: malignancy with TIV	Unequivocal enhancing soft tissue in vein, regardless of visualization of parenchymal mass	Multidisciplinary discussion for tailored workup May include biopsy

**Table 2 cancers-15-05118-t002:** Staging Systems for HCC.

	Primary Tumor (T)	Regional Lymph Nodes (N)	Metastases (M)
AJCC TNM [57]			
T1a	Solitary tumor ≤ 2 cm	Nx, regional lymph nodes cannot be assessed	M0, no distant metastasis
T1b	Solitary tumor ≥ 2 cm without vascular invasion	N0, no regional lymph node metastasis	M1, distant metastasis
T2	Solitary tumor > 2 cm with vascular invasion, or multiple tumors, none > 5 cm	N1, regional lymph node metastasis	
T3	Multiple tumors at least one of which is >5 cm		
T4	Single tumor or multiple tumors of any size involving a major branch of the portal vein or hepatic vein or tumor(s) with direction invasion of adjacent organs other than the gallbladder or with perforation of visceral peritoneum		
Stage			
Stage IA	T1a	N0	M0
Stage IB	T1b	N0	M0
Stage II	T2	N0	M0
Stage IIIA	T3	N0	M0
Stage IIIB	T4	N0	M0
Stage IVA	Any T	N1	M0
Stage IVB	Any T	Any N	M1
Histologic Grade (G)		Fibrosis Score (F)	
Gx	Grade cannot be accessed	F0	Fibrosis score 0–4 (none to moderate fibrosis)
G1	Well differentiated	F1	Fibrosis score 5–6 (severe fibrosis or cirrhosis)
G2	Moderately differentiated		
G3	Poorly differentiated		
G4	Undifferentiated		
BCLC (47)			
BCLC-0 (very early stage)	Single ≤ 2 cm Preserved liver function, PS 0		
BCLC-A (early stage)	Single, or ≤3 nodules, all ≤3 cm Preserved liver function, PS 0		
BCLC-B (intermediate stage)	Multinodular Preserved liver function, PS 0		
BCLC-C (advanced stage)	Portal invasion and/or extrahepatic spread Preserved liver function, PS 12		
BCLC-D (terminal stage)	Any tumor burden End-stage liver function, PS 34		

**Table 3 cancers-15-05118-t003:** Approved systemic therapies for HCC.

Study Name	Treatment	Inhibited Molecules	Dose
First-Line Therapies			
IMbrave150 [122]	Atezolizumab plus bevacizumab	PD-L1 (immune checkpoint), VEGF (angiogenesis)	Atezolizumab: 1200 mg IV every 3 weeks Bevacizumab: 15 mg/kg IV every 3 weeks
HIMALAYA [124,126]	Durvalumab + tremelimumab	PD1 and CTLA4 (immune checkpoints)	Tremelimumab 300 mg IV infusion + durvalumab 1500 mg IV
SHARP [116]	Sorafenib	VEGFR, PDGFR (angiogenesis), MAPK (BRAF)	400 mg oral twice daily
REFLECT [117]	Lenvatinib	VEGFR, PDGFR, FGFR (angiogenesis), KIT, RET	12 mg oral once daily, if bodyweight ≥ 60 kg 8 mg oral once daily, if bodyweight < 60 kg
Second-line therapies			
RESOURCE [127]	Regorafenib	VEGFR, PEDGR (angiogenesis), MAPK (BRAF)	160 mg orally once daily on days 1–21 of each 28-day cycle
CELESTIAL [128]	Cabozantinib	MET (proliferation), VEGFR (angiogenesis), RET	60 mg orally once daily
REACH-2 [129]	Ramucirumab	VEGFR2 (angiogenesis)	8 mg/kg IV every 2 weeks
KEYNOTE-240 [130]	Pembrolizumab	PD1 (immune checkpoint)	200 mg IV every 3 weeks
KEYNOTE-224 [131]	Pembrolizumab	PD1 (immune checkpoint)	200 mg every 3 weeks for ≤35 cycles
CheckMate 040 [132]	Nivolumab + ipilimumab	PD1 and CTLA4 (immune checkpoints)	Nivolumab 1 mg/kg IV + ipilimumab 3 mg/kg IV every 3 weeks for four cycles, Nivolumab 240 mg IV every 2 weeks
RATIONALE-301 [133,134]	Tislelizumab	PD1 (immune checkpoint)	200 mg IV once every three weeks
CARES-310 [135]	Camrelizumab + rivoceranib (apatinib)	PD1 (immune checkpoint) VEGFR2 (angiogenesis)	Camrelizumab 200 mg IV every two weeks and rivoceranib 250 mg tablet orally once daily

**Table 4 cancers-15-05118-t004:** Ongoing clinical trials in systemic therapy for HCC.

Study Name	Treatment	Inhibited Molecules	Primary Endpoint(s)	Dose
COSMIC-312 [136,137] NCT03755791	Cabozantinib + atezolizumab versus sorafenib And Cabozantinib versus sorafenib	Cabozantinib is a multikinase inhibitor. Atezolizumab is an immune checkpoint inhibitor.	PFS, OS	Cabozantinib 40 mg oral, qd + atezolizumab 1200 mg infusion, q3w versus sorafenib 400 mg twice daily.
LEAP-002 [138] NCT03713593	Lenvatinib + pembrolizumab versus lenvatinib + placebo	Lenvatinib targets VEGFR2-3, FGFR1-2, RET, PDGFR. Pembrolizumab is an anti-PD-1 antibody.	PFS, OS	Levatinib 12 mg (for participants with screening body weight ≥60 kg) or 8 mg (for participants with screening body weight <60 kg) orally once a day + pembrolizumab 200 mg by intravenous infusion on day 1 of each 21-day cycle (administered for up to 35 cycles). Lenvatinib 12 mg (for participants with screening body weight ≥60 kg) or 8 mg (for participants with screening body weight <60 kg) orally once a day plus saline placebo by IV infusion on day 1 of each 21-day cycle.
CheckMate 9DW [139] NCT04039607	Nivolumab + ipilimumab versus sorafenib or lenvatinib	Nivolumab is an anti-PD1 receptor antibody. Ipilimumab is an anti-CTLA-4 antibody. Sorafenib is a multikinase (RAF1, BRAF, VEGFR, −1, −2, −3, PDGFR, KIT, FGFR1, RET) inhibitor inhibiting cell proliferation and angiogenesis.	OS	Nivolumab IV infusion + ipilimumab IV infusion versus sorafenib oral tablet or Lenvatinib oral tablet.
GOING [140] NCT04170556	Regorafenib (monotherapy for the first 8 weeks) + nivolumab	Regorafenib potently blocks multiple protein kinases involved in tumor angiogenesis (VEGFR, −2, −3, TIE2), oncogenesis (KIT, RET, RAF-1, BRAF, BRAFV600E), metastasis (VEGFR3, PDGFR, FGFR) and tumor immunity (CDF1R). Nivolumab is a IgG4 monoclonal antibody to (PD)-1 receptor.	Safety	Regorafenib 160 mg/day for 3 weeks on and 1 week off + nivolumab 1.5 mg/kg, 3 mg/kg, or 240 mg/infusion every 2 weeks.
N/A [141] NCT04183088	Part 1: regorafenib + tislelizumab Part 2: regorafenib + tislelizumab versus regorafenib	Regorafenib potently blocks multiple protein kinases involved in tumor angiogenesis (VEGFR, −2, −3, TIE2), oncogenesis (KIT, RET, RAF-1, BRAF, BRAFV600E), metastasis (VEGFR3, PDGFR, FGFR), and tumor immunity (CDF1R). Tislelizumab is an anti-PD-1 antibody.	Part 1: safety Part 2: PFS, ORR	Part 1: Tislelizumab 200 mg IV on day 1 every 3 weeks + regorafenib orally 80 mg per day. Part 2 (group 1): receives tislelizumab 200 mg IV on day 1 + regorafenib (dosage in randomized cohort will be determined according to results in the safety cohort). Part 2 (group 2): regorafenib 80 mg daily for 1 week, regorafenib 120 mg daily for week 2, regorafenib 160 mg daily for week 3, dosing-free interval for week 4.
REVERT [142] NCT05440708	TTI-101 And TTI-101 + pembrolizumab And TTI-101 + atezolizumab + bevacizumab	TTI-101 is a STAT3 inhibitor. Pembrolizumab is a PD-1 inhibitor. Atezolizumab is a PD-L1 inhibitor. Bevacizumab targets angiogenesis by inhibiting VEGF.	Phase 1: safety, MTD, RP2D Phase 2: ORR	Phase 1: participants will receive up to 3 dose levels of TTI-101 oral tablet as single agent to determine RP2D. Then treated in Phase 2 with TTI-101 RP2D as single agent. TTI-101 oral tablet up to 3 dose levels + Pembrolizumab IV infusion to determine RP2D. Then in Phase 2 treat with RP2D of TTI-101 + pembrolizumab. TTI-101 oral tablet up to 3 dose levels to determine RP2D + atezolizumab IV infusion + bevacizumab IV infusion. Then in Phase 2 treat with RP2D of TTI-101 in combination with atezolizumab + bevacizumab.
ARTEMIS [143] NCT04797884	TheraBionic Device versus Placebo Device	TheraBionic Device emits emitting radiofrequencies programmed with hepatocellular carcinoma-specific modulation frequencies.	OS, quality of life	TheraBionic Device emits hepatocellular carcinoma-specific modulation frequencies >200 for three courses of 60-min treatments of modulated radiofrequencies >200 administered in morning, noon, and evening for 6 weeks (very first treatment administered at recruiting site, all others at home) versus Placebo that does not emit any hepatocellular carcinoma-specific modulation frequencies for three courses of 60-min treatments administered morning, noon, and evening for 6 weeks (very first treatment administered at recruiting site, all others at home).
N/A [144] NCT02715362	Vascular interventional therapy mediated GPC3-targeted chimeric antigen receptor T (CART) cells	Patient’s autologous T cells are activated and engineered to express chimeric antigen receptors (CARs) specific for GPC3, expanded, and returned to patient.	Safety	Transcatheter arterial infusion of: (1–10) × 10^6^ CAR positive T cells/kg. A single dose of 1.5 g/m^2^ of cyclophosphamide will be given two days before CART cell infusion.
N/A [145] NCT03993743	CD147-targeted CART cells by hepatic artery infusion	Patient’s autologous T cells are activated and engineered to express chimeric antigen receptors (CARs) specific for CD147, expanded, and returned to patient.	Safety	Three CD147-CART doses infused by hepatic artery at 1-week intervals.
N/A [146] NCT05323201	fhB7H3-targeted CART cells by transhepatic arterial infusion	Patient’s autologous T cells are activated and engineered to express chimeric antigen receptors (CARs) specific for B7H3, expanded, and returned to patient.	Safety, ORR	Before infusion, cyclophosphamide (750 mg/m^2^ IV) and fludarabine (30 mg/m^2^ IV) will be administered for three consecutive days. Two days after lymphodepletion, fhB7H3 CART cells will be infused by transhepatic arterial infusion at three dose levels (1 × 10^6^/kg, 3 × 10^6^/kg, and 5 × 10^6^/kg) only one time.

**Table 5 cancers-15-05118-t005:** Image response criteria.

	Complete Response (CR)	Partial Response (PR)	Stable Disease (SD)	Progressive Disease (PD)
WHO [165]	Disappearance, confirmed at 4 weeks	50% decrease in measurable lesions; confirmed at 4 weeks	Neither PR nor PD criteria met	25% increase in measurable lesions; no CR, PR, or SD documented before increased disease
RECIST [166]	Disappearance; confirmed at 4 weeks	30% decrease in target lesions; confirmed at 4 weeks	Neither PR nor PD criteria met	20% increase in sum of diameter of target lesions; no CR, PR, or SD documented before increased disease
EASL [168]	Disappearance; determined by two observations not less than 4 weeks apart	50% decrease in maximum diameter of the enhanced tumor area	Neither PR nor PD criteria met	25% increase in sum of diameter of the enhanced tumor area
qEASL [167,169]	Disappearance	65% decrease in enhanced tumor volume	Neither PR nor PD criteria	73% increase in the enhanced tumor volume
mRECIST [167,169,170]	Disappearance of any intratumoral arterial enhancement in all target lesions	30% decrease in the sum of diameters of viable (enhancement in the arterial phase) target lesions	Neither PR nor PD criteria met	20% increase in the sum of viable target lesions
vRECIST [167,169]	Disappearance	65% decrease in enhanced tumor volume	Neither PR nor PD criteria	73% increase in the enhanced tumor volume

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
