# Peer review of "Hepatocellular Carcinoma: Surveillance, Diagnosis, Evaluation and Management"

_cancers, 2023, doi:10.3390/cancers15215118_

Round 1

Reviewer 1 Report

Well written and comprehensive review. My comments:

THe authors should comment on the subsets of population needing HCC screening, with particular regard to recently identified risk factors for HCC, such as obesity (cite the recent MA: PMID: 33721336 )

The authors should comment the main response criteria used in hepato-oncology (EASL, WHO, RECIST, mRECIST)

The authors should comment on the topic of eventual adjuvant therapies after curative treatments, citing the results of the STORM trial and the other drugs testing in that setting (cite the series PMID: 25974743)

Any inference on liver transplant is missing in the paper

A table with ongoing trials in the setting of systemic therapy would be useful

Reviewer 2 Report

The titled presented here is very interesting topic. But the revision of the manuscript is required. This study is focused on extensive review on HCC. There are so  many articles are missing in the review process. As an expert opinion, I suggest that either include maximum studies related to HCC diagnosis or detection. Author has to also mention that the selection process of article. Author has to clearly mention that the which article is qualify for the inclusion or not qualify for the inclusion in the study. The image quality is not in adequate resolution. So requesting that please provide the adequate quality image. the orientation of Figure 2 is not proper so do it in proper.

Proof read required.

Round 2

Reviewer 1 Report

The revised version is OK. Thank you!